# Serum Immunoglobulin Changes in Multiple Myeloma Patients Treated with CAR T-Cell Therapy

**DOI:** 10.3390/cimb47080640

**Published:** 2025-08-09

**Authors:** Alexa Burger, Ulrike Bacher, Michele Hoffmann, Katja Seipel, Christof Schild, Inna Shaforostova, Thomas Pabst

**Affiliations:** 1Department of Medical Oncology, Inselspital, University of Bern, CH-3010 Bern, Switzerland; alexa.burger@students.unibe.ch (A.B.); michele.hoffmann@insel.ch (M.H.); innaivanovna.shafosrostova@insel.ch (I.S.); 2Department of Hematology, Inselspital, University of Bern, CH-3010 Bern, Switzerland; veraulrike.bacher@insel.ch; 3Department for Biomedical Research (DBMR), University of Bern, CH-3008 Bern, Switzerland; katja.seipel@unibe.ch; 4Department of Clinical Chemistry, Inselspital, University of Bern, CH-3010 Bern, Switzerland

**Keywords:** multiple myeloma (MM), CAR T-cell therapy, isotype switching, immuno-fixation electrophoresis (IFE)

## Abstract

Chimeric antigen receptor (CAR) T-cell therapy has emerged as a promising treatment for relapsed or refractory multiple myeloma (RRMM), with high response rates of 80–95%. Serum immunoglobulin changes have been observed throughout conventional multiple myeloma treatment, including after immunomodulatory drugs, proteasome inhibitors, and autologous stem cell transplantation. However, the clinical significance of new abnormal protein bands (APBs) following CAR T-cell therapy is largely unexplored. We retrospectively analyzed consecutive multiple myeloma (MM) patients who received CAR T-cell therapy at the University Hospital Bern between May 2021 and February 2024. Serum paraprotein (M-protein) patterns were assessed using immuno-fixation electrophoresis (IFE) before and after CAR T-cell treatment. Patients were grouped based on serum immunoglobulin changes. Among 46 patients, 9 (19.6%) developed new APBs following CAR T-cell therapy. No significant differences in overall survival (OS) or progression-free survival (PFS) were observed between patients with and without APBs. Immunoglobulin changes occurred in both relapsed and non-relapsed patients, suggesting that the appearance of new APBs does not indicate disease progression. This observation aligns with previous reports of paraprotein changes following conventional MM therapies. This report suggests that new APBs following CAR T-cell therapy are a relatively common finding but do not correlate with inferior clinical outcomes. Our results highlight the need for larger, multi-center studies to further investigate this phenomenon in MM patients undergoing CAR T-cell therapy.

## 1. Introduction

Despite significant progress in the treatment of MM in recent decades, the disease remains mostly incurable following standard treatment options and is marked by the presence of multiple sub-clones, reflecting significant tumor heterogeneity. The malignant plasma cells produce monoclonal immunoglobulins or their components (light and/or heavy chains), known as paraproteins [1]. The excessive proliferation of plasma cells can lead to organ damage, commonly presenting with one or more of the following clinical features: hypercalcemia, renal dysfunction, anemia, or bone lesions [2]. 

Chimeric antigen receptor (CAR) T-cell therapy has become a promising treatment approach for patients with relapsed or refractory multiple myeloma (RRMM). This therapy has demonstrated high overall response rates, ranging from 64% to 88% [3,4,5,6]. CAR T-cell therapy contributes to the activation and regulation of the immune system. By engineering T cells to specifically recognize and attack tumor cells, CAR T-cell therapy can exert significant anti-myeloma immune responses [7]. 

Following high-dose chemotherapy (HDCT) and autologous stem cell transplantation (ASCT), the transient appearance of abnormal protein bands (APBs) or paraprotein switching on serum immuno-fixation electrophoresis has been observed in up to 73% of patients [8,9]. However, rates are lower after other conventional therapies. In myeloma patients receiving cytotoxic agents or novel induction regimens, oligo-clonal bands occurred in 33%, with a higher incidence in novel agent-treated patients (60%) than in those receiving cytotoxic therapy alone (11.1%) [10]. Similarly, immunomodulatory therapy led to an atypical serum immuno-fixation pattern in 33% of cases [11]. High-dose chemotherapy has been proposed to disrupt normal T-cell regulation of B-cell proliferation, impair B-cell affinity maturation, and hinder the antigenic recall response. Such alterations in humoral immunity may lead to a reactivation of early B-cell developmental pathways, manifesting as the appearance of APBs. Consequently, the presence of APBs may reflect the restoration of B-cell function following chemotherapy or CAR T-cell therapy and may serve as a marker of immune reconstitution [12]. The emergence of APBs has been associated with a favorable prognosis, particularly in patients undergoing HDCT/ASCT [12,13,14], but has also been reported after allogenic or syngeneic transplantation [15] and immunomodulatory therapies [11].

However, our understanding of the prevalence and clinical significance of APBs and paraprotein changes in serum immuno-fixation electrophoresis (IFE) following CAR T-cell therapy remains limited. To date, one report describes the APB phenomenon, which was observed in four patients with RRMM treated with anti-BCMA CAR T cells [13]. Furthermore, a case report describes the occurrence of multiple immunoglobulin isotype switches following CAR T-cell therapy [12]. Remarkably, in this case, a stringent complete response was maintained for over three years without the need for additional treatment. 

To better understand the occurrence of APBs and paraprotein changes after CAR T-cell therapy, we retrospectively analyzed myeloma patients treated with CAR T-cell therapy at the University Hospital Bern for changes in immunoglobulins at follow-up independently of reoccurrence or response.

## 2. Materials and Methods

### 2.1. Patients and Clinical Data

A single-center, retrospective cohort study was conducted, including RRMM patients treated with CAR T-cell therapy at the University Hospital Bern between May 2021 and February 2024. The treatment regimen included standard lympho-depleting chemotherapy including fludarabine and cyclophosphamide, followed by CAR T-cell infusion. All patients received the CAR T cell product Idecabtagene Vicleucel (Abecma). Patients were followed up until November 2024. Written informed consent was obtained from all patients, and this study adhered to the principles of the Declaration of Helsinki. Patient data were obtained from electronic health records and laboratory databases of the University Hospital Bern. Variables collected included demographics, paraprotein subtypes, stage of MM at diagnosis, MM diagnostic criteria, cytogenetics, remission status, and prior lines of therapy. Immunoglobulin patterns were assessed at diagnosis, prior to CAR T-cell therapy, and during follow-up. Based on the IFE findings after CAR T-cell therapy, patients were categorized into three groups: (1) those who developed new APBs, (2) those with no detectable M-proteins, and (3) those who retained the same M-protein pattern as before therapy. The absence of detectable M-protein in IFE reflected either a complete response or early relapse without the emergence of detectable bands. In patients with APBs, the specific nature of these alterations was analyzed carefully. A comparison was made between relapsed and non-relapsed patients. Diagnostic and efficacy criteria were applied based on the guidelines established by the International Myeloma Working Group.

### 2.2. Paraprotein Analysis

Serum para-protein patterns were analyzed using immuno-fixation electrophoresis (IFE) utilizing the HYDRAGEL 4 IF kit (Sebia, Norcross, GA, USA), and M-proteins were quantified by capillary zone electrophoresis and densitometry on Capillarys 3 Octa (Sebia). Serum free light chains (FLCs) were measured by the N Latex FLC assay (Siemens, Munich, Germany). An APB was defined, based on IFE findings, as a new or altered para-protein pattern not matching the original monoclonal spike. These changes involved alterations in heavy or light chain type and the appearance of new bands not previously present. This included complete changes in heavy chain type, or additions/losses of immunoglobulin components following therapy. In one case, appearance of the involved monoclonal light chain in serum IFE without the corresponding intact M-protein was suggestive of a light chain escape. This was confirmed by a quantitative serum FLC assay. 

### 2.3. Minimal Residual Disease 

Minimal residual disease (MRD) was evaluated using multicolor flow cytometry as previously described [14].

### 2.4. Statistical Analysis

Statistical analysis and survival curves were calculated using GraphPad Prism software version 10 as previously described [14]. 

## 3. Results

### 3.1. Baseline Clinical Characteristics of the MM Patient Cohort

Forty-six patients with RRMM who received CAR T-cell therapy were included in this study. Detailed patient characteristics are summarized in Table 1. At diagnosis, the median age of the cohort was 60 years (range: 40–78), with a male-to-female ratio of 2.3 (32 males and 14 females). The distribution of para-protein subtypes was as follows: 58.7% IgG, 23.9% IgA, and 2.2% IgM, with 32.6% having kappa light chain involvement and 50.0% lambda light chain involvement. Additionally, 17.4% of patients were diagnosed with light chain-only disease. Based on the Revised International Staging System (R-ISS), 26.1% of patients were classified as Stage I, 43.5% as Stage II, and 28.3% as Stage III. Diagnostic criteria were predominantly marked by anemia (89.1%) and osteolytic lesions (78.3%). Cytogenetic data were available for 86.9% of the patients, with 13.0% exhibiting high-risk t(4;14), 19.6% del(17p), and 21.7% gain(1q). The median time from initial diagnosis to CAR T-cell therapy was 62 months (range: 11–205). Regarding treatment history, 82.6% of patients had undergone prior HDCT/ASCT. At the time of CAR T-cell infusion, 47.8% of patients had progressive disease, 23.9% stable disease, and 23.9% partial responses. 

### 3.2. Association of Abnormal Protein Bands and Clinical Relapse

The relapse rate following CAR T-cell therapy was slightly higher in patients without APBs (59.5%) compared to those with APBs (44.4%). Detailed clinical and laboratory characteristics of patients at remission or at relapse are summarized in Table 2. Among the 26 patients who experienced clinical relapse, 4 (15.4%) had APBs, 19 patients (73%) were paraprotein-positive with the same immunoglobulin pattern, and 3 (11.5%) showed no evidence of a para-protein in IFE (Table 3). Notably, these three patients died early after CAR T-cell therapy and were therefore classified as having early relapse/progression due to rapid clinical deterioration before biochemical reappearance of a para-protein. In contrast, among the 20 patients who did not experience clinical relapse, 5 (25%) developed APBs, 3 (15%) retained the same para-protein, and 12 (60%) no detectable para-protein. 

### 3.3. Characteristics of Abnormal Protein Bands in Relapsed vs. Non-Relapsed Patients

Different para-protein alterations were observed after CAR T-cell therapy, including the appearance of new APBs (20%), maintenance of same pattern (48%), and loss of bands (32%) (Table 3). APB protein quantities were mostly maintained during follow-up, including several cases of rising and few cases of decreasing quantities. Of the 37 patients without APBs, 22 patients (47.8%) still showed a detectable para-protein by IFE but retained the same pattern as before CAR T-cell therapy, whereas 15 (32.6%) no longer exhibited any para-protein in IFE. Patients with APBs were more frequently classified as R-ISS Stage II (66.7%) than those without (37.8%). t(4;14) was more prevalent among patients with APBs (33.3%) than those without (8.1%). The majority of relapsed patients maintained the same para-protein pattern, while the majority of the non-relapsed patients had a loss of M-protein. Among the nine patients who developed APBs, distinct patterns of para-protein alterations were observed, including the appearance of new APBs without the initial M-protein, light chain escape, and changes in heavy chain components (both additions of APBs and losses). In the subgroup of non-relapsed patients (n = 5), two patients (patient #1 and #8) showed new APBs without the initial M-protein. In patient #1, the initial biclonal pattern IgA kappa (>10 g/L) plus IgG kappa disappeared from IFE after CAR T, and new APBs, IgM kappa and IgG lambda, appeared. In patient #8, the initial M-protein IgG lambda (>10 g/L) disappeared from IFE after CAR T and a new APB IgG Kappa appeared. The remaining three non-relapsed patients (patients #4, #7, and #9) showed additions of APBs (IgG kappa and IgG lambda). Notably, appearance of new APBs without the initial M-protein was observed only in non-relapsed patients. In contrast, the relapsed patients (n = 6) exhibited a more heterogeneous set of changes. Patient #2 experienced light chain escape, transitioning from IgG kappa to LC kappa. Patients #3 and #8 both lost one intact M-protein from a biclonal pattern after CAR T, while patient #5 acquired a new IgG lambda APB. Specific characteristics of the nine patients with immunoglobulin alterations are detailed in Table 4.

### 3.4. No Association of Abnormal Protein Bands and Survival Outcomes

Myeloma patients treated with CAR T-cell therapy in this study had a median survival of 20 months, ranging from <1 to >42 months. There were no significant differences in progression-free (PFS) or overall survival (OS) in patients with and without APBs (Figure 1).

## 4. Discussion

In this study, we investigated the occurrence of APBs and serum immunoglobulin changes in patients with multiple myeloma (MM) following CAR T-cell therapy. Both relapse and non-relapse cases were analyzed during the follow-up period. Prior studies have addressed the emergence of APBs or para-protein changes in the context of CAR T-cell therapy, but data on this phenomenon remain scarce. Li et al. reported a case of RRMM treated with anti-BCMA CAR T-cell therapy, where the patient underwent three distinct immunoglobulin class switches while maintaining a stringent complete response for over three years [12]. 

A recent study investigating BCMA-CD38 bispecific CAR T-cell therapy in patients with RRMM demonstrated high response rates and manageable cytokine release syndrome (CRS). Notably, the study also observed a transient clonal isotype switch (CIS) after CAR T-cell infusion, which was classified as a benign phenomenon, with no evidence of disease progression [15]. Similarly, a case report further demonstrated isotype switching from IgA lambda to IgG lambda within six months post-CAR T-cell therapy in a 66-year-old patient, who remained asymptomatic for three years [16]. In another clinical trial of BCMA CAR T-cell therapy by Liang et al., APBs were identified in four patients, none of whom showed symptoms or evidence of disease recurrence at the time of detection [13]. To the best of our knowledge, this study represents the first attempt to systematically analyze APBs in both relapsed and non-relapsed MM patients after CAR T-cell therapy, providing new insights into this phenomenon and its clinical relevance. 

All patients in our study received Idecabtagene Vicleucel (Idecel). Response rates following CAR T-cell therapy showed a complete response (CR) in 32 patients, very good partial response (VGPR) in 3 patients, and partial response (PR) in 3 patients, resulting in an overall response rate (ORR) of 82.6% (38 out of 46 patients). This ORR is slightly higher than the 79% ORR reported in the international multi-center study by Merz et al. [17]. In comparison, another study has reported a response rate of 73%, and a systematic review and meta-analysis by Roex et al. with 640 patients treated with BCMA-targeted CAR T-cell therapy showed an ORR of 80.5% [18,19]. While this study specifically analyzed the effects of Idecel, previous research has suggested that different CAR T-cell products exhibit distinct cellular expansion dynamics [17]. These differences may influence immune reconstitution and subsequent immunoglobulin changes.

The median time to the appearance of APBs following CAR T-cell infusion was 7 months (range: 2–17 months). Notably, in one patient, the newly detected immunoglobulins disappeared again within 3 months. In the remaining eight patients, the immunoglobulins persisted throughout the follow-up period. In comparison, the report by Liang et al. suggested a median time to occurrence of APBs at 4 months post-CAR T-cell infusion, with a median duration of 5 months until they disappeared again [13]. 

We observed that 9 of 46 CAR T-cell therapy patients (19.6%) developed APBs after treatment, suggesting that this phenomenon may not be uncommon in this cohort. This aligns with findings from Cho et al., who observed clonal isotype switching in 31.7% of patients after ASCT, and Ye et al., who reported a similar occurrence of 22% in ASCT recipients. Both studies associated isotype switching with favorable outcomes, including improved PFS and OS [20,21]. Liang et al. also reported APBs in 33% of CAR T-cell therapy patients and observed a higher complete response rate and better outcomes in these patients compared to those without APBs [12]. This supports the notion that immunoglobulin changes are more likely a benign and transient phenomenon. A recent retrospective study by Liu et al. reported that 48.9% of patients developed APBs. Similar to the findings of Liang et al., they observed that the presence of APBs was associated with significantly improved responses to Cita-cel therapy, as well as longer overall survival (OS) and progression-free survival (PFS) [22]. 

Lu et al., however, investigated paraprotein changes specifically in relapsed patients, independent of the therapy type. They suggested that complete isotype switching at relapse, as well as light chain escape, was a rare event, occurring in only 1.6% of patients in both groups combined [1]. In contrast, our study identified a notably higher prevalence of APBs, with 23.9% of CAR T-cell therapy patients affected—six patients (23%) in the relapse group and five patients (25%) in the non-relapse group. This discrepancy may be attributed to differences in study populations, treatment approaches, or follow-up durations. Furthermore, Lu et al. focused on serum immunoglobulin changes only at relapse, whereas our study assessed their occurrence in the post-treatment phase independently of relapse. The 4 relapsed patients and 5 non-relapsed patients who developed APBs suggest that, even within a small study population of 46 patients, serum immunoglobulin changes are not associated with disease relapse or the emergence of malignant plasma cell clones. We found no evidence that the development of APBs after CAR T-cell therapy was associated with inferior outcomes. There were no differences in the PFS between the relapse and non-relapse groups (Log Rank *p*-value = 0.87) as well as in the OS between the groups (Log Rank *p*-value = 0.95). Studies investigating APB occurrence after ASCT in MM have demonstrated correlations with improved survival outcomes. However, Kaplan–Meier analysis in this study revealed no prognostic significance in the post-CAR T-cell setting, as neither OS nor PFS was improved in patients with APBs. 

Interestingly, patients with APBs exhibited distinct clinical and biological features. These patients had higher incidences of anemia, hypercalcemia, and elevated creatinine levels. Additionally, they demonstrated a greater prevalence of Revised International Staging System (R-ISS) Stage II disease and cytogenetic abnormalities, such as t(4;14), which are associated with adverse outcomes in MM. Despite the differences, both groups had comparable median ages and similar frequencies of anemia, hypercalcemia, renal impairment, and extra-medullary disease. Liang et al. also found similar baseline characteristics in patients with and without APBs [13]. While these findings suggest that specific subsets of patients may be predisposed to developing APBs, the underlying mechanisms driving this phenomenon remain unclear. Several mechanisms may account for the occurrence of APBs. One is the ‘intra-clonal class switch’ hypothesis, which suggests that unknown molecular genetic events promote the Ig-class switch. Another is the different progenitor B-cell lines’ hypothesis, which postulates that pressure from BM38 CAR-T cells temporarily disrupts the balance between B cells and Ig-producing plasma cells, leading to the transient occurrence of APBs [23,24]. Liang et al. hypothesize that the mechanism of para-protein changes may be similar to that observed after high-dose chemotherapy [13]. The appearance of APBs could be linked to the reconstitution of the immune system and the recovery of B-cell function, following the reduction in myeloma-induced immunosuppressive effects after high-dose chemotherapy [25,26]. All referenced studies reporting abnormal protein bands are summarized in Table 5. 

While our study did not specifically assess infection rates, the persistent hypo-gamma-globulinemia described by Wang et al. suggests that APBs may be part of a broader immune reconstitution process rather than a direct indicator of disease relapse [27]. Future research should explore whether certain immunoglobulin shifts correlate with increased infection susceptibility post-CAR T-cell therapy. 

This study has several limitations. Most importantly, the APBs observed may originate from different biological sources. They could arise from a sub-clone of the original MM (representing true isotype switching), from a secondary plasma cell clone such as secondary MGUS, or from a transient immunological response to therapy. Without molecular characterization, their exact origin remains uncertain, which complicates interpretation. Furthermore, the clinical impact of such bands remains speculative without confirmatory molecular analysis (e.g., sequencing, flow cytometry). It is essential to carefully assess para-proteins after CAR T therapy to differentiate true isotype switching from apparent switches, as well as consider the dynamics of these changes for a more accurate prognostic evaluation. Additionally, this was a single-center study with a relatively small sample size (n = 46), which limits the detection of subtle associations. Due to the small number of patients in the APB+ group, our study was only sufficiently powered to detect relatively large differences in outcomes between groups. The retrospective design inherently introduces potential biases, including selection bias and incomplete data collection. These factors may affect the generalizability of our findings to broader and more diverse patient populations. The follow-up period of up to 42 months restricts conclusions about the long-term clinical implications of APBs and serum immunoglobulin changes, particularly their relevance to survival outcomes and relapse dynamics. Finally, the absence of a control group treated with alternative therapies limits comparisons across different treatment approaches.

## 5. Conclusions

Appearance of abnormal protein bands (APBs) following CAR T-cell therapy in patients with MM is a frequent event, and in this cohort, we found that 19.6% of patients had APBs. While APBs are common, they are not necessarily associated with disease relapse or worse outcomes in terms of PFS or OS. Therefore, changes in immunoglobulin profiles should not be interpreted as relapse without considering the patient’s overall clinical status. Despite the limitations of this study, we believe it provides valuable insights and highlights the need for prospective, multi-center trials to further explore the clinical significance of APBs in the context of CAR T-cell therapy. Future research should focus on the mechanisms underlying APBs, including potential molecular genetic events and the role of immune system reconstitution. Additionally, larger patient cohorts, extended follow-up periods, and comparisons across various treatment modalities are essential to confirm the findings and assess whether immunoglobulin changes could serve as a predictive biomarker in the management of multiple myeloma patients receiving CAR T-cell therapy.

## Figures and Tables

**Figure 1 cimb-47-00640-f001:**
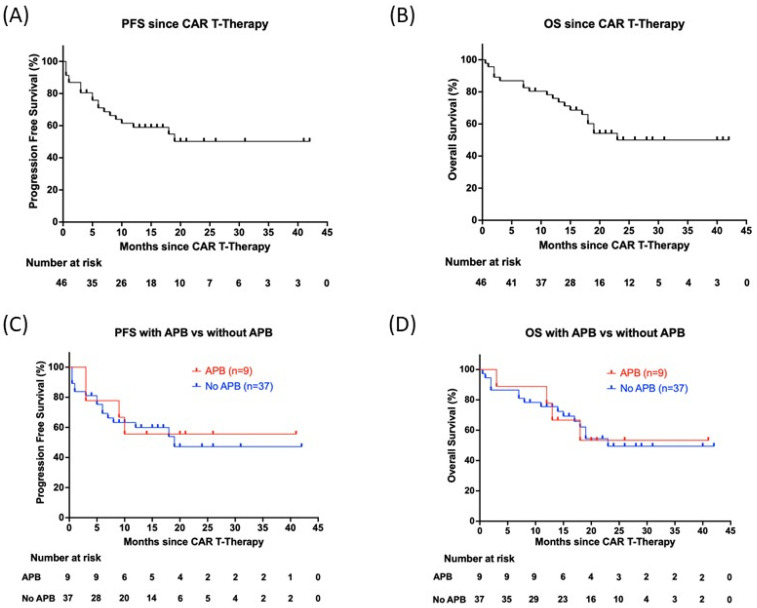
Progression-free survival and overall survival of the entire patient cohort (n = 46). (**A**) PFS entire cohort; (**B**) OS entire cohort; (**C**) PFS in patients with APBs (n = 9) and without APBs (n = 37), *p*-value 0.87; (**D**) OS in patients with and without APBs, *p*-value 0.95.

**Table 1 cimb-47-00640-t001:** Patient characteristics at initial diagnosis.

Parameter	All Pts (n = 46)	Pts w APBs (n = 9)	Pts w/o APBs (n = 37)	*p* *
Median age, years (range)	60 (40–78)	59 (44–75)	60 (40–78)	
Males/females, n (ratio)	32/14 (2.3)	8/1 (8)	24/13 (1.8)	
M-protein subtype, n (%)				
Kappa light chain	15 (32.6%)	1 (11.1%)	6 (16.2%)	1
Lambda light chain	23 (50.0%)	0 (0%)	6 (16.2%)	0.327
IgG	27 (58.7%)	5 (55.6%)	22 (59.5%)	1
IgA	11 (23.9%)	5 (55.6%)	6 (16.2%)	0.025
IgM	1 (2.2%)	0 (0.0%)	1 (2.7%)	1
Light chain only	8 (17.4%)	0 (0.0%)	8 (21.6%)	0.324
R-ISS stage, n (%)				
Available	45 (97.8%)	9 (100%)	36 (97.3%)	
I	12 (26.1%)	1 (11.1%)	11 (29.7%)	0.409
II	20 (43.5%)	6 (66.7%)	14 (37.8%)	0.149
III	13 (28.3%)	2 (22.2%)	11 (29.7%)	1
MM diagnostic criteria				
Anemia (Hb, f < 121 g/L, m < 135 g/L), n (%)	41 (89.1%)	8 (88.9%)	33 (89.2%)	1
Hypercalcemia (>2.55 mmol/L), n (%)	14 (30.4%)	3 (33.3%)	11 (29.7%)	1
Creatinine (f > 84 mmol/L, m > 104 mmol/L)	19 (41.3%)	3 (33.3%)	16 (43.2%)	0.716
Osteolytic lesions, n (%)	36 (78.3%)	8 (88.9%)	28 (75.7%)	0.659
Extramedullary disease, n (%)	5 (8.7%)	2 (22.2%)	3 (8.1%)	0.248
Cytogenetics				
Available, n (%)	40 (86.9%)	8 (88.9%)	32 (86.5%)	
t(4;14) translocation, n (%)	6 (13.0%)	3 (33.3%)	3 (8.1%)	0.079
Del(17p), n (%)	9 (19.6%)	1 (11.1%)	8 (21.6%)	0.664
Gain(1q), n (%)	10 (21.7%)	2 (22.2%)	8 (21.6%)	1
No high-risk cytogenetics, n (%)	15 (32.6%)	3 (33.3%)	12 (32.4%)	1

Patients (Pts), abnormal protein bands (APBs), * *p*-value Fisher’s exact test.

**Table 2 cimb-47-00640-t002:** Patient characteristics at the time of CAR T-cell therapy.

Parameter	All Pts (n = 46)	Pts w APBs (n = 9)	Pts w/o APBs (n = 37)	*p* *
Time ID to CAR T, months, median (range)	62 (11–205)	86 (21–200)	61 (11–205)	
Remission status				1
Progressive disease, n (%)	22 (47.8%)	4 (44.4%)	18 (48.6%)	
Stable disease, n (%)	11 (23.9%)	2 (22.2%)	9 (24.3%)	
Partial response, n (%)	11 (23.9%)	2 (22.2%)	9 (24.3%)	
Very good partial response, n (%)	1 (2.2%)	0 (0.0%)	1 (2.7%)	
Complete response, n (%)	1 (2.2%)	1 (11.1%)	0 (0.0%)	
Prior lines of therapy, median (range)	5 (2–11)	6 (2–7)	5 (2–11)	1
Previous HDCT/ASCT				1
Yes, n (%)	38 (82.6%)	8 (88.9%)	30 (81.1%)	
No, n (%)	8 (17.4%)	1 (11.1%)	7 (18.9%)	
Clinical Relapse after CAR T-cell therapy				0.472
Yes, n (%)	26 (56.5%)	4 (44.4%)	22 (59.5%)	
No, n (%)	20 (43.5%)	5 (55.6%)	15 (40.5%)	

Initial diagnosis (ID), patients (Pts), abnormal protein bands (APBs), * *p*-value Fisher’s exact test.

**Table 3 cimb-47-00640-t003:** Serum immunoglobulin changes after CAR T-cell therapy.

Parameter	All Pts (n = 46)	Pts w Relapse (n = 26)	Pts w/o Relapse (n = 20)	*p* *
Abnormal Protein Bands	9 (19.6%)	4 (15.4%)	5 (25%)	0.472
New APB without initial M-protein	2 (4.3%)	0 (0%)	2 (10%)	0.183
Light chain escape	1 (2.2%)	1 (3.8%)	0 (0%)	1.0
New APB addition alongside initial M-protein	4 (8.7%)	1 (3.8%)	3 (15%)	0.303
Loss of one intact M-protein from biclonal	2 (4.3%)	2 (7.7%)	0 (0%)	0.497
No Abnormal Protein Bands	37 (80.4%)	22 (84.6%)	15 (75%)	0.472
No more M-protein in IFE	15 (32.6%)	3 (11.5%)	12 (60%)	**0.0011**
Same isotype as before	22 (47.8%)	19 (73%)	3 (15%)	**0.0001**

Patients (Pts), abnormal protein bands (APBs), * *p*-value Fisher’s exact test.

**Table 4 cimb-47-00640-t004:** Characteristics of myeloma patients with immunoglobulin isotype changes following CAR T-cell therapy.

#	Age at ID	Sex	Primary Paraprotein	ISS Stage	FISH	BR Pre CART	Relapse	Paraprotein APB	BR Post CART	PFS (mo)	OS (mo)	FLC-Ratio	ΔFLC mg/L
**1**	58	m	IgG Kappa, IgA Kappa	II	t(4;14)	PD	No	IgM Kappa, IgG Lambda	CR	41	41	1	0.1
**2**	44	m	IgG Kappa	II	t(4;14), gain1q	PD	Yes	LC Kappa	PR	3	3	3840	1920
**3**	45	m	IgG Kappa, LC Lambda	II	del(17p)	SD	Yes	LC Lambda	CR	3	18	<0.01	332
**4**	49	m	IgG Lambda	I	normal	PD	No	IgG Lambda, IgG Kappa	VGPR	26	26	2	0.5
**5**	65	m	IgA Lambda	II	gain 1q	CR	Yes	IgA Lambda, IgG Lambda	CR	9	13	0.04	47.8
**6**	62	m	IgG Lambda	III	normal	PR	No	IgG Kappa	CR	21	21	1	0.5
**7**	67	m	IgG Kappa	II	t(4:14)	SD	No	IgG Kappa, IgG Lambda	CR	20	20	1.2	1.9
**8**	75	m	IgA Kappa, IgG Kappa	III	n.d.	PD	Yes	IgA Kappa	CR	10	12	18.3	58.9
**9**	60	f	LC Lambda	II	normal	PR	No	IgG Lambda, IgG Kappa	CR	14	14	0.17	92

ID: initial diagnosis; ISS: international staging system; FISH: fluorescent in situ hybridization; BR: best response; PFS: progression free survival; OS: overall survival; FLC: free-light chain; CR: complete response; PR: partial response; VGPR: very good partial response, PD: progressive disease, SD: stable disease.

**Table 5 cimb-47-00640-t005:** Reference studies reporting abnormal protein bands (APBs).

Author(s)	Year	Ref.	Study Type	Synopsis/Key Findings	Impact of APBs on Outcome
Lu et al.	2023	[1]	Retrospective (various therapies)	Isotype switch in 3.5% of relapsed patients (complete switching, light chain escape, non-secretory relapse)	No significant difference in prognosis of patients
Maisnar et al.	2007	[8]	Retrospective (HDCT/ASCT)	APBs in 43% of patients post ASCT, most likely due to immune reconstitution, not disease relapse	Associated with improved outcome
Hall et al.	2009	[9]	Retrospective (HDCT/ASCT)	APBs in 73% of patients, representing regeneration of a limited immune response	Favorable event-free survival
Fernández de Larrea et al.	2011	[10]	Retrospective (conventional and novel therapies)	APBs in 33.3% of patients, and in 60% of patients with novel agents	No assessment of impact on outcome
Mark et al.	2008	[11]	Retrospective (BiRD regimen)	Atypical IFE patterns in 33% likely reflect robust antitumor response and immune reconstitution	Significantly greater OR and CR
Ye et al.	2019	[20]	Retrospective (HDCT/ASCT)	Clonal isotype switching in 22% of ASCT patients, suggestive of more profound T-cell immune reconstitution	Improved PFS and OS
Liang et al.	2022	[13]	Case series (CAR T-cell, n = 12)	APBs in 33% likely reflect immune reconstitution	Favorable prognosis
Cho et al.	2025	[21]	Prospective (HDCT/ASCT)	Clonal isotype switch in 31.7% of patients, reflecting robust immune reconstitution	Favorable outcomes
Liu et al. (LEGEND-2)	2024	[22]	Retrospective (CAR T-cell)	48.9% developed APBs post-CAR T, indicating robust immune reconstitution	Longer PFS and OS, higher response rates
Zent et al.	1998	[25]	Retrospective (HDCT/ASCT)	APBs in 10% of patients likely reflect immune recovery	Significantly higher CR, longer OS

CR: complete response; OR: overall response; PFS: progression-free survival; OS: overall survival.

## Data Availability

No data supporting the reported results are deposited elsewhere.

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
