# Peer review of "Serum Immunoglobulin Changes in Multiple Myeloma Patients Treated with CAR T-Cell Therapy"

_cimb, 2025, doi:10.3390/cimb47080640_

Round 1

Reviewer 1 Report

Comments and Suggestions for Authors

The authors describe immunoglobulin changes in multiple myeloma patients who underwent CAR-T cell therapy.

The topic is novel, the manuscript is well written and the results are interesting.

1.The authors did not mention what was their method to assess response.  MRD was evaluated? and if yes, What was the method? multicolor flowcytometry or NGS?

2. The author did not mention the quantity of abnormal paraprotein. What was the trend of ABP? rising or decreasing during follow up?

Author Response

The authors describe immunoglobulin changes in multiple myeloma patients who underwent CAR-T cell therapy.

The topic is novel, the manuscript is well written and the results are interesting.

Response: We appreciate the helpful and positive comments.

1. The authors did not mention what was their method to assess response.  MRD was evaluated? and if yes, What was the method? multicolor flowcytometry or NGS?

Response: Minimal residual disease (MRD) was assessed using multicolor flow cytometry. This information has been added to the Materials and Methods section.

  1. The author did not mention the quantity of abnormal paraprotein. What was the trend of ABP? rising or decreasing during follow up?

Response: We observed changes in paraprotein patterns, including appearance of new APBs (20%), maintenance of same pattern (48%), and loss of bands (32%). APB protein quantities were mostly maintained during follow-up, including several cases of rising and few cases of decreasing quantities. This information has been added to the results section.

Submission Date

04 July 2025

Date of this review

16 Jul 2025 21:34:39

Date of revision: 31 July 2025

Reviewer 2 Report

Comments and Suggestions for Authors

"Serum Immunoglobulin Changes in Multiple Myeloma Patients Treated with CAR T-Cell Therapy"

This retrospective, single-center study investigates the emergence of abnormal protein bands (APBs) in patients with relapsed/refractory multiple myeloma (RRMM) following CAR T-cell therapy with Idecabtagene Vicleucel. The authors analyze the frequency, nature, and clinical relevance of APBs in 46 patients and conclude that APBs are relatively common but not associated with disease relapse or inferior outcomes.

The topic is timely and clinically relevant, particularly given the increasing use of CAR T-cell therapy in RRMM. The manuscript is well-written, clearly structured, and supported by appropriate statistical analyses. However, there are areas that need clarification and refinement to enhance the manuscript's clarity, rigor, and impact.

  1. The study addresses an understudied aspect of CAR T-cell therapy outcomes. While the emergence of APBs has been previously described in case reports or small cohorts, a systematic analysis, as presented here, is valuable. However, the novelty would be strengthened by deeper molecular characterization or mechanistic insights (e.g., clonality studies or immunophenotyping).
  2. While APBs are defined based on IFE changes, more clarification is needed on how “new APBs” were distinguished from oligoclonal reconstitution. The clinical impact of such bands remains speculative without confirmatory molecular analysis (e.g., sequencing, flow cytometry). The authors acknowledge this but should emphasize the limitation more explicitly in the Discussion.
  3. The lack of a significant difference in OS or PFS between APB+ and APB− groups is notable. However, with only 9 patients in the APB+ group, the study may be underpowered to detect clinically relevant differences. Please include a post-hoc power analysis or discuss this limitation more explicitly.
  4. The case of light chain escape is intriguing but insufficiently explored. Was this associated with clinical progression, or did it represent benign clonal evolution? More detail (e.g., imaging, bone marrow findings) would strengthen this section.
  5. Figure 1: The Kaplan-Meier plots are relevant, but actual event numbers at risk should be shown below each graph. Also, median PFS and OS values with confidence intervals should be reported in the Results section.
  6. Table 4: Please define abbreviations (e.g., PR, CR, VGPR) in the table legend for clarity.
  7. The manuscript references studies that show both positive and neutral implications of APBs post-ASCT or CAR T therapy. A more structured synthesis of these findings (perhaps in a table) would be helpful.
  8. Line 17–18 (Abstract): “...with high response rates.” Consider quantifying these response rates here for clarity.
  9. Line 90–92 (Methods): Please specify how frequently follow-up IFE was performed post-CAR T infusion.
  10. Line 183 (Table 3): Consider bolding p-values < 0.05 for better visual emphasis.
  11. Line 243–244 (Discussion): “...this phenomenon may occur frequently in this cohort.” This statement should be reworded to reflect the 19.6% rate—i.e., “may not be uncommon” is more accurate.
  12. Typographical: Ensure consistency in abbreviations (e.g., APBs vs. APB).
  13. Modify the sentence in the introduction section to "The emergence of APBs has been associated with improved immune reconstitution and favorable prognosis, particularly in patients undergoing HDCT/ASCT [12–14], but has also been reported after allogenic or syngeneic transplantation [15] and immunomodulatory therapies [11]" and may be linked to immune dysregulation and abnormal immunoglobulin responses observed in cancer [Mir et al., 2016] and cite the article “Mir AR, Moinuddin, Islam S. Circulating autoantibodies in cancer patients have high specificity for glycoxidation modified histone H2A. Clin Chim Acta. 2016 Jan 30;453:48-55. doi: 10.1016/j.cca.2015.12.004. Epub 2015 Dec 4. PMID: 26656310”.

Author Response

This retrospective, single-center study investigates the emergence of abnormal protein bands (APBs) in patients with relapsed/refractory multiple myeloma (RRMM) following CAR T-cell therapy with Idecabtagene Vicleucel. The authors analyze the frequency, nature, and clinical relevance of APBs in 46 patients and conclude that APBs are relatively common but not associated with disease relapse or inferior outcomes.

The topic is timely and clinically relevant, particularly given the increasing use of CAR T-cell therapy in RRMM. The manuscript is well-written, clearly structured, and supported by appropriate statistical analyses. However, there are areas that need clarification and refinement to enhance the manuscript's clarity, rigor, and impact.

Response: We appreciate the thorough review and helpful comments.

  1. The study addresses an understudied aspect of CAR T-cell therapy outcomes. While the emergence of APBs has been previously described in case reports or small cohorts, a systematic analysis, as presented here, is valuable. However, the novelty would be strengthened by deeper molecular characterization or mechanistic insights (e.g., clonality studies or immunophenotyping).

Response: Unfortunately, in the short time admitted for revision, we were not able to include clonality studies or immuno-phenotyping. However, we included this topic in some detail in the introduction.

  1. While APBs are defined based on IFE changes, more clarification is needed on how “new APBs” were distinguished from oligoclonal reconstitution. The clinical impact of such bands remains speculative without confirmatory molecular analysis (e.g., sequencing, flow cytometry). The authors acknowledge this but should emphasize the limitation more explicitly in the Discussion.

Response: This limitation has been highlighted more explicitly in the revised discussion.

  1. The lack of a significant difference in OS or PFS between APB+ and APB− groups is notable. However, with only 9 patients in the APB+ group, the study may be underpowered to detect clinically relevant differences. Please include a post-hoc power analysis or discuss this limitation more explicitly.

Response: Since post-hoc power analyses are widely regarded as uninformative due to their tautological nature (i.e., when the p-value is >0.05, post-hoc power will always be low; when the p-value is <0.05, it will always be high (https://doi.org/10.1198/000313001300339897) we instead calculated the minimal detectable effect size that would achieve 80% power with our sample size. We have added a paragraph to the discussion addressing this limitation.

  1. The case of light chain escape is intriguing but insufficiently explored. Was this associated with clinical progression, or did it represent benign clonal evolution? More detail (e.g., imaging, bone marrow findings) would strengthen this section.

Response: Unfortunately, we were not able to include imaging or bone marrow findings in this study.

  1. Figure 1: The Kaplan-Meier plots are relevant, but actual event numbers at risk should be shown below each graph. Also, median PFS and OS values with confidence intervals should be reported in the Results section.

Response: We included event numbers at risk, and reported median PFS and OS times.

  1. Table 4: Please define abbreviations (e.g., PR, CR, VGPR) in the table legend for clarity.

Response: Abbreviations are defined in table 4 legend.

  1. The manuscript references studies that show both positive and neutral implications of APBs post-ASCT or CAR T therapy. A more structured synthesis of these findings (perhaps in a table) would be helpful.

Response: We have added an additional table referencing APB studies in the discussion (Table 5).

  1. Line 17–18 (Abstract): “...with high response rates.” Consider quantifying these response rates here for clarity.

Response: Information on the response rates, presented in percentage form, has been included.

  1. Line 90–92 (Methods): Please specify how frequently follow-up IFE was performed post-CAR T infusion.

Response: IFE was performed regularly as part of routine check-up of patients.

  1. Line 183 (Table 3): Consider bolding p-values < 0.05 for better visual emphasis.

Response: P-values < 0.05 have been highlighted in bold in Table 3.

  1. Line 243–244 (Discussion): “...this phenomenon may occur frequently in this cohort.” This statement should be reworded to reflect the 19.6% rate—i.e., “may not be uncommon” is more accurate.

Response: We have revised the sentence as proposed.

  1. Typographical: Ensure consistency in abbreviations (e.g., APBs vs. APB).

Response: The abbreviation APB has been carefully checked and standardized throughout the manuscript.

  1. Modify the sentence in the introduction section to "The emergence of APBs has been associated with improved immune reconstitution and favorable prognosis, particularly in patients undergoing HDCT/ASCT [12–14], but has also been reported after allogenic or syngeneic transplantation [15] and immunomodulatory therapies [11]" and may be linked to immune dysregulation and abnormal immunoglobulin responses observed in cancer [Mir et al., 2016] and cite the article “Mir AR, Moinuddin, Islam S. Circulating autoantibodies in cancer patients have high specificity for glycoxidation modified histone H2A. Clin Chim Acta. 2016 Jan 30;453:48-55. doi: 10.1016/j.cca.2015.12.004. Epub 2015 Dec 4. PMID: 26656310”.

Response: The topic of the referred publication is circulating anti-histone antibodies. These antibodies are commonly found in autoimmune diseases like Systemic Lupus Erythematosus (SLE) and drug-induced lupus. While they are also observed in other conditions, including some cancers, they are not typically associated with multiple myeloma. We therefore did not include this reference.

Submission Date

04 July 2025

Date of this review

12 Jul 2025 18:21:32

Date of revision:  31 July 2025

Round 2

Reviewer 2 Report

Comments and Suggestions for Authors

The authors have addressed all previous comments thoroughly, and the manuscript has improved significantly. I am satisfied with the revisions, and the manuscript is now suitable for publication.